# ERL-MR: Harnessing the Power of Euler Feature Representations for Balanced Multi-modal Learning

Weixiang Han
School of Data Science and Technology, Heilongjiang University
Harbin, China
weixianghan@s.hlju.edu.cn

Chengjun Cai
Department of computer science, City University of Hong Kong (Dongguan)
Dongguan, China
chengjun.cai@cityu-dg.edu.cn

Yu Guo
School of Artificial Intelligence, Beijing Normal University
Beijing, China
yuguo@bnu.edu.cn

Jialiang Peng*
School of Data Science and Technology, Heilongjiang University
Harbin, China
jialiangpeng@hlju.edu.cn

## Abstract

Multi-modal learning leverages data from diverse perceptual media to obtain enriched representations, thereby empowering machine learning models to complete more complex tasks. However, recent research results indicate that multi-modal learning still suffers from "*modality imbalance*": *Certain modalities' contributions are suppressed by dominant ones, consequently constraining the overall performance enhancement of multimodal learning.* To tackle this issue, current approaches attempt to mitigate modality competition in various ways, but their effectiveness is still limited. To this end, we propose an **E**uler **R**epresentation **L**earning-based **M**odality **R**ebalance (ERL-MR) strategy, which reshapes the underlying competitive relationships between modalities into mutually reinforcing win-win situations while maintaining stable feature optimization directions. Specifically, ERL-MR employs Euler's formula to map original features to complex space, constructing cooperatively enhanced non-redundant features for each modality, which helps reverse the situation of modality competition. Moreover, to counteract the performance degradation resulting from optimization drift among modalities, we propose a Multi-Modal Constrained (MMC) loss based on cosine similarity of complex feature phase and cross-entropy loss of individual modalities, guiding the optimization direction of the fusion network. Extensive experiments conducted on four multi-modal multimedia datasets and two task-specific multi-modal multimedia datasets demonstrate the superiority of our ERL-MR strategy over state-of-the-art baselines, achieving modality rebalancing and further performance improvements.

## CCS Concepts

• **Computing methodologies → Artificial intelligence**.

*Corresponding author.

## Keywords

Multi-modal Learning, Modality Imbalance, Euler Formula, Multi-modal Constrained Loss

**ACM Reference Format:**
Weixiang Han, Chengjun Cai, Yu Guo, and Jialiang Peng. 2024. ERL-MR: Harnessing the Power of Euler Feature Representations for Balanced Multi-modal Learning. In *Proceedings of the 32nd ACM International Conference on Multimedia (MM '24), October 28–November 1, 2024, Melbourne, VIC, Australia.* ACM, New York, NY, USA, 10 pages. https://doi.org/10.1145/3664647.3681215

## 1 Introduction

The advent of the *multi-modal learning* paradigm [35, 43, 50] represents a pivotal advancement in artificial intelligence (AI) technology, elevating AI models to a new level of capability [7, 39, 58]. This paradigm empowers AI models with rich perceptual abilities, enabling them to undertake increasingly complex multimedia tasks, including sentiment analysis [14, 79], audio-visual speech recognition [32, 78], and visual question answering [29], among others. Specifically, multi-modal learning endeavors to mine and analyze data from multiple sensory media, harnessing insights from diverse sources. This approach is widely recognized for its potential to enhance AI model performance in comparison to training with uni-modal data.

However, recent research results [9, 16, 40, 60, 63, 68, 78] have revealed a counterintuitive fact: *the learning processes within each modality of multi-modal learning exhibit a dynamic competition rather than the anticipated mutual benefit.* As an illustration, Huang *et al.* in [16] devised a theoretical analytical framework to substantiate the aforementioned claims, subsequently yielding theoretical insights. Their research underscores that in some cases, the optimal uni-modal network outperforms the multi-modal network trained jointly, a finding which challenges conventional wisdom [35, 43, 50]. In this context, researchers attribute this phenomenon to the "*modality imbalance*" issues [9, 40, 68], wherein a dominant modality restricts the comprehensive exploitation of the multiple modalities at play. This, in turn, undermines the fundamental objective of enhancing model performance through the fusion of information from diverse modalities.

To further explain the reasons behind the above phenomena, a series of empirical and theoretical methods [9, 15, 16, 24, 40, 62, 63, 68] have been proposed to analyze the learning process of multi-modal learning. Some of them try to explain it from the perspective of

joint training, that is, they believe that different modalities will compete with each other during the process of joint training the fusion network via gradient descent [15, 16]. More specifically, the competitive behavior materializes as the dominant modality converges rapidly with a substantial gradient amplitude, effectively impeding the convergence and optimization of the weaker modality [1, 10]. Another predominant perspective underscores the challenge posed by the heterogeneity of multi-modal data, specifically concerning the acquisition of knowledge related to multi-modal correlations and complementarities [9, 40, 63]. These modal correlations and complementarities are manifested through feature representations. However, heterogeneous feature representations can disrupt intermodal learning and result in incongruent learning efficiencies across different modalities.

In light of the aforementioned insights, researchers are dedicated to the development of multi-modal learning strategies aimed at addressing the challenge of "modality imbalance." To tackle the issue, prior works employ gradient-based methods, including gradient angle constraints [68], on-the-fly gradient modulation [40], and gradient balance constraints [63]. While these approaches effectively regulate the learning rates of various modalities through gradient adjustments and constraints, they may still fall short in preventing the dominant modality from interfering with the gradient update direction of the weaker modality [9, 15, 16]. This limitation poses a challenge in enhancing the performance of the slow-learning modality. Another set of methods concentrates on formulating efficient feature utilization techniques designed to mitigate the negative effects of heterogeneous multi-modal data on learning speed [3, 9, 22, 30, 38, 68, 70]. The key idea of their approach revolves around energizing the slower learning modality through facilitating more robust feature utilization, while concurrently mitigating the influence of the dominant modality by slowing down itself during the early stages of training. Clearly, the solely focusing on enhancing feature utilization and learning efficiency of the weaker modality may not yield benefits for the fusion network, as they impede the learning process of the dominant modality.

**Current Dilemma.** The prevailing dilemma manifests itself as a delicate trade-off. When attempts are made to mitigate modality imbalance by restraining the optimization of dominant modalities, the result can be a decline in the performance of these strong modalities, consequently exerting a negative impact on the overall performance of the fusion model [9, 16]. On the other hand, if efforts are oriented toward stimulating slower-learning modalities to foster modality balance, it might entail a deceleration in the optimization pace of the dominant modalities, consequently diminishing the advantages they bring to the fusion network [15]. *Hence, the key challenge lies in finding a solution to break through this dilemma and devise a way that can simultaneously tackle both issues effectively.*

**Our Contributions.** To address the aforementioned challenges, we explore a new path, one that eschews the traditional strategies of either suppressing dominant modalities or invigorating weaker ones to achieve modality balance and boost fusion network performance. Specifically, we propose an **E**uler **R**epresentation **L**earning-based **M**odality **R**ebalance (ERL-MR) strategy. In this strategy, our insight centers on the concept that the Euler vector space transformation of modal features can aptly capture the interplay between modalities, effectively reshaping their relationship from competition to mutual

reinforcement. Furthermore, our observations indicate that imposing constraints on the feature phases between different modalities and optimizing the cross-entropy loss for each modality proves advantageous in enhancing both modality-specific performance and the overall performance of fusion network. Therefore, we have made the following designs:

*(D1.)* – **Euler Transformation Design.** In this design, ERL-MR endeavors to transform the vector feature representations of diverse modalities into a complex vector space using *Euler's formula*, with the primary objective of enhancing modal features. Subsequently, we harness the geometric attributes of the Euler transformation to effectively reconfigure the relationship between modalities, transitioning it from competition to mutual reinforcement. The reason behind this approach is that within the complex space, our design has the capacity to model the relationships of phase and modulus associated with different modalities, thereby facilitating the integration of feature interactions among the modalities.

*(D2.)* – **Constrained Loss Design.** In this design, ERL-MR aims to take into account the performance of each modality and the fusion model. To achieve this, we propose a joint loss, the multi-modal constrained loss function, including two critical components: the cosine similarity constraint, aligning optimization directions between modalities, and the cross-entropy loss constraint, indicating the correct optimization direction for each modality. In particular, in the cosine similarity constraint, we constrain the cosine value of the phase of the complex features between each modality, which is equivalent to performing a geometric transformation on complex features.

The main contributions are summarized below:

- We propose an effective modality rebalance strategy, *i.e.*, ERL-MR, against modal imbalance in multi-modal learning. This strategy is model-agnostic and can naturally be integrated into the framework of multi-modal learning.
- We leverage insights from multi-modal imbalance analysis to propose two simple yet effective designs: Euler transformation and multi-modal loss. Additionally, we provide a complexity analysis of the overall strategy.
- We conduct extensive case studies on six real-world multi-modal datasets, demonstrating that our strategy significantly outperforms existing state-of-the-art methods.

## 2 Related Work

### 2.1 Multi-modal Learning

Multi-modal learning aims to extract complementary or independent knowledge from various modalities, enabling the representation of multi-modal data [2, 28, 71]. This endeavor empowers AI models with the capability to comprehend and process diverse modal information [65]. Within this domain, a valuable research direction is how to efficiently extract and fuse meaningful representations [25, 31, 72]. For example, researchers leverage graph representations of multi-modal multimedia data to employ knowledge graph embedding techniques [6, 54, 61], facilitating the accomplishment of diverse complex tasks, *e.g.*, biological knowledge graph completion [66, 67], knowledge graph reasoning [75], and graph recommendation systems [42, 76, 77]. Furthermore, multi-modal learning technique can serve as a foundational building block

within a model to harness the potential of multi-modal information, leading to enhanced model performance in specific domains, *e.g.*, traffic trajectory prediction [47, 48, 69], disease diagnosis [4], action recognition [5, 11, 33], audio-visual speech recognition[32], and visual question answering [29]. In recent times, multi-modal techniques have gained significant prominence within the domain of large language models (LLM), resulting in the development of multi-modal LLMs tailored for specific applications. Notable examples include the HuaTuo medical large model [56], Llama 2 [53], GPT-4 vision model [36, 46], and others, each designed for specific fields. However, as mentioned above, how to harness the power of each modality data in multi-modal learning is still an open question.

## 2.2 Imbalanced Multi-modal Learning

**Existing Rebalance Approaches and Their Limitations.** One of the straightforward reasons hindering the realization of the full potential of each modality data is the issue of modality imbalance [5]. To address the problem of modality imbalance, researchers have devised a variety of techniques that can be categorized into two main groups: gradient-based methods [40, 60, 63, 68] and feature-based methods [3, 9, 38, 78]. For gradient-based methods, their insight is to suppress the convergence speed of the dominant modality through gradient modulation or constraints to achieve modality balance. Nevertheless, these methods may entail a reduction in the dominant mode's contribution to the fusion network, which is not conducive to enhancing the overall performance of the fusion network [9]. On the other hand, feature-based methods, as employed in prior research, concentrate on extracting the features of weaker modes to expedite their convergence for achieving modal balance. However, such methods do not change the nature of competition between modalities and cannot significantly improve the overall performance of the fusion network [16]. Hence, existing methods tend to address only a limited aspect of the problem, as they do not alter the fundamental competitive relationship between modalities, which prevents them from mutually benefiting each other.

## 2.3 Euler Representation Learning

Euler representation learning is a machine learning technique designed to develop feature representation methods capable of directly processing geometric data, such as 3D shapes and graphics [27, 34, 57, 59, 64, 74]. Researchers have delved into approaches involving Euler feature representations to capture the inherent geometric characteristics of the data, enabling neural networks (*e.g.*, graph neural networks (GNN) [8, 13, 23]) to effectively handle geometry-related tasks. For example, Jiang *et al.* in [18] applied Euler features to characterize the geometric correlations among users on the Taobao platform. They harnessed the potent learning capabilities of GNN to craft a recommendation system capable of accommodating millions of users. Hence, Euler representation is frequently employed in the realm of representation learning and the learning of geometric topological structures, primarily due to its advantageous properties for feature enhancement and geometric feature transformation [3, 17, 26, 38, 49]. To this end, we aim toward balanced multi-modal learning by leveraging the special power of Euler feature representation in this regard.

## 3 Preliminaries

### 3.1 Modality Imbalance

Consider a multi-modal dataset $\mathcal{D}$, which includes $M$ distinct modalities, labeled as $m_0, m_1, \ldots, m_{M-1}$. In the Multi-modal learning, encoders $\psi^{m_1}(\omega^{m_0}, \cdot), \psi^{m_1}(\omega^{m_1}, \cdot), \ldots, \psi^{m_1}(\omega^{m_{M-1}}, \cdot)$ to extract features from modality $m_0, m_1, \ldots, m_{M-1}$, respectively, where $\omega$ is the parameters of the encoder. Let $f$ represent the fusion model that has been trained on the dataset $\mathcal{D}$. In this context, the performance of the multi-modal fusion model and each modality can be denoted as $f(\mathcal{D})$ and $f(m_0), f(m_1), \ldots, f(m_{M-1})$, respectively. When the performance of a specific modality significantly exceeds that of other modalities, this situation is termed modality imbalance, with the particular modality $m_i$ ($i \in \{1, 2, \cdots, M-1\}$) identified as the dominant modality. For each modality $m_i$, during each iteration $t$, if $m_i$ is the dominant modality, the performance measure $f(m_i)$ will converge more rapidly compared to the other modalities. In addition, we have provided theoretical proof for the definition of *Modality Imbalance* [9]. The complete proof can be found in the full version.

### 3.2 Euler's Formula

We briefly introduce the formal expression of Euler's formula and its application in complex vector spaces [52]. Euler's formula is a foundational mathematical equation in complex analysis, establishing a fundamental connection between trigonometric functions and the complex exponential function. Specifically, for any real number $x$, Euler's formula states:

$$e^{ix} = \cos x + i \sin x, \tag{1}$$

where $e$ is the base of the natural logarithm and $i$ is the imaginary unit. Eq. (1) can also be formulated as:

$$\lambda e^{i\theta} = \underbrace{\lambda \cos \theta}_{real} + i(\underbrace{\lambda \sin \theta}_{imaginary}), \tag{2}$$

where $\lambda e^{i\theta}$ and $\lambda \cos \theta + i(\lambda \sin \theta)$ represent the complex vector in polar and rectangular forms, respectively. Here, $\lambda$ and $\theta$ are the modulus and phase of the complex vector. For a complex vector $r + ip$, we define the real part as $r = \lambda \cos \theta$ and the imaginary part as $p = \lambda \sin \theta$. The modulus $\lambda$ and phase $\theta$ can be expressed as follows:

$$\lambda = \sqrt{r^2 + p^2}, \theta = \text{atan2}(p, r) \tag{3}$$

where $\text{atan2}(\cdot, \cdot)$ denotes the 2-argument arctangent function. The application of Euler's formula enables the conversion of complex vectors from the rectangular form to the polar form, facilitating the encoding of features in the polar space.

## 4 Methodology

In contrast to prior efforts that sought to address this problem by constraining the optimization speed of dominant modalities or hastening the training of weaker modalities, our objective is to delve into the intricate relationships between different modality features with the aim of establishing a state of mutual reinforcement. To this end, we design two components in the ERL-MR strategy: **Euler Feature Transformation** and **Multi-modal Constrained Loss**. The overview of the proposed strategy is shown in Fig. 1.

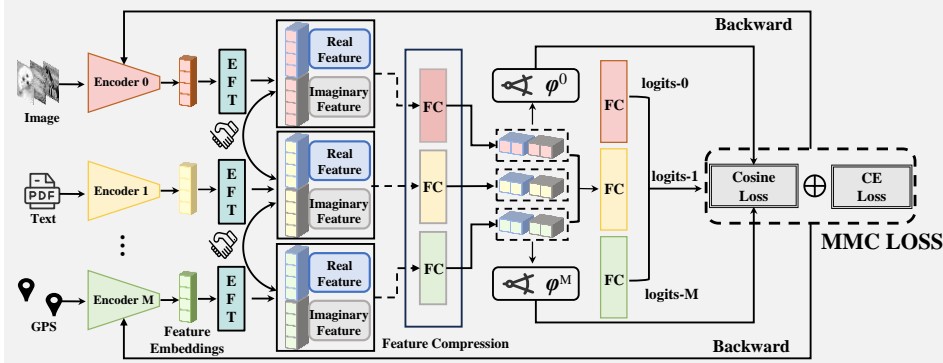

**Figure 1: Overview of ERL-MR strategy, where EFT and FC denote the Euler feature transformation and the fully connected layer, respectively.**

## 4.1 Feature Mapping via Euler Feature Transformation

**Complex Space Mapping.** Our key insight revolves around the likelihood that the phenomenon of modality imbalance may be attributed to the heterogeneity within the feature spaces of different modalities. In this context, the first problem we face is how to map the heterogeneous feature space to a homogeneous feature space to alleviate the imbalance phenomenon. Inspired by Euler's formula (*i.e.*, Eq. (2)), we aim to use it to implement feature mapping. Specifically, we map the feature embedding vectors produced by the modal encoder $\psi$ from real vector space to complex vector space via Euler's formula. Consider a feature embedding $\boldsymbol{e}_j = \psi_j^{m_i}(\omega^{m_i}, x_i^{m_i})_{j=1}^N$ produced by the encoder of modality $m_i$. Following Eq. (2), we can map $\psi_j^{m_i}(\omega^{m_i}, x_i^{m_i})$ into the complex vector space, representing it with two real vectors $\boldsymbol{r}$ and $\boldsymbol{p}$ to denote the real and imaginary parts of the complex vector, respectively. Thus, we have:

$$\tilde{\boldsymbol{e}}_j = \boldsymbol{r}_j + i\boldsymbol{p}_j, \tilde{\boldsymbol{e}}_j = \tilde{\psi}_j^{m_i}(\omega^{m_i}, x^{m_i}), \quad (4)$$

where $\tilde{\boldsymbol{e}}_j$ is a feature embedding in the complex vector space. To transform feature embeddings into complex vectors, the fundamental concept is to regard them as *phases* and integrate learnable parameters $\mu_j$ as moduli (*i.e.*, $\mu_j \rightarrow \boldsymbol{\lambda}, \boldsymbol{e}_j \rightarrow \boldsymbol{\theta}$), following Euler's formula as presented in Eq. (2):

$$\widetilde{\psi}_j^{m_i}(\omega^{m_i}, x^{m_i}) = \underbrace{\mu_j^{m_i} \cos\left(\psi_j^{m_i}(\omega^{m_i}, x^{m_i})\right)}_{real}$$
$$+ i \underbrace{\mu_j^{m_i} \sin\left(\psi_j^{m_i}(\omega^{m_i}, x^{m_i})\right)}_{imaginary}. \quad (5)$$

For simplicity, we let:

$$\psi_{\boldsymbol{r}}^{m_i}(\omega^{m_i}, x^{m_i}) \triangleq \{\mu_j^{m_i} \cos\left(\psi^{m_i}(\omega^{m_i}, x^{m_i})\right)\}_{j=1}^N$$
$$\psi_{\boldsymbol{p}}^{m_i}(\omega^{m_i}, x^{m_i}) \triangleq \{\mu_j^{m_i} \sin\left(\psi^{m_i}(\omega^{m_i}, x^{m_i})\right)\}_{j=1}^N, \quad (6)$$

where $N$ is the number of features. Thus, in order to simplify the expression, we can rewrite Eq. (5) as follows:

$$\widetilde{\psi}_j^{m_i}(\omega^{m_i}, x^{m_i}) = \psi_{\boldsymbol{r}}^{m_i}(\omega^{m_i}, x^{m_i}) + i\psi_{\boldsymbol{p}}^{m_i}(\omega^{m_i}, x^{m_i}). \quad (7)$$

**Insight 1.** In multi-modal feature representation learning, Euler's formula maps features to the complex vector space, preserving and

sharing valuable information among different modalities. Moreover, the sine and cosine operations inherent in Euler's formula contribute to alleviating feature differences between different modalities by employing polar coordinate transformation. Therefore, our method aims at enhancing feature presentation and minimizing feature presentation disparities, effectively transforming the competitive relationship between modalities into a cooperative one.

**Feature Compression.** Although complex space mappings offer feature-enhanced representations, their integration into the model architecture introduces challenges, notably increased computational costs. This aspect poses difficulties in the practical deployment of this strategy. To address this issue, we design a compression operation that uses a simple linear layer to compress the features obtained by the complex space mapping operation. Its formal definition is as follows:

$$\hat{\psi}_{\boldsymbol{r}}^{m_i}(\omega^{m_i}, x^{m_i}) = W_{compress}^{m_i} \cdot \left[\psi_{\boldsymbol{r}}^{m_i}(\omega^{m_i}, x^{m_i})\right]$$
$$+ b_{compress}^{m_i},$$
$$\hat{\psi}_{\boldsymbol{p}}^{m_i}(\omega^{m_i}, x^{m_i}) = W_{compress}^{m_i} \cdot \left[\psi_{\boldsymbol{p}}^{m_i}(\omega^{m_i}, x^{m_i})\right] \quad (8)$$
$$+ b_{compress}^{m_i},$$

where $\hat{\psi}_{\boldsymbol{r}}^{m_i}(\omega^{m_i}, x^{m_i})$ and $\hat{\psi}_{\boldsymbol{p}}^{m_i}(\omega^{m_i}, x^{m_i})$ respectively represent the real part feature and imaginary part feature obtained after the compression operation, $W_{compress}^{m_i} \in \mathbb{R}^{N \times \left(d_{\psi_{\boldsymbol{r}}^{m_i}(\omega^{m_i}, x^{m_i})}\right)}$ and $b_{compress}^{m_i} \in \mathbb{R}^{d_{\psi_{\boldsymbol{r}}^{m_i}(\omega^{m_i}, x^{m_i})}}$ represent the parameters of the linear layer. After compressing the real and imaginary features, we apply a nonlinear activation function, *i.e.*, ReLU, to generate the compressed feature representations via Eq. (9).

$$\psi_{new_{\boldsymbol{r}}}^{m_i}(\omega^{m_i}, x^{m_i}) = \text{ReLU}\left(\hat{\psi}_{\boldsymbol{r}}^{m_i}(\omega^{m_i}, x^{m_i})\right),$$
$$\psi_{new_{\boldsymbol{p}}}^{m_i}(\omega^{m_i}, x^{m_i}) = \text{ReLU}\left(\hat{\psi}_{\boldsymbol{p}}^{m_i}(\omega^{m_i}, x^{m_i})\right). \quad (9)$$

Subsequently, we execute a concatenation operation using Eq. (10) to merge the real and imaginary features into new features for model training.

$$\psi_{\text{New}}^{m_i}(\omega^{m_i}, x^{m_i}) = \psi_{new_{\boldsymbol{r}}}^{m_i}(\omega^{m_i}, x^{m_i}) \oplus$$
$$\psi_{new_{\boldsymbol{p}}}^{m_i}(\omega^{m_i}, x^{m_i}), \quad (10)$$

**Table 1: Summary of the six multi-modal datasets.**

| Dataset | Modality Information | # Classes | Object | Domain | # Samples | Size (MB) |
|---|---|---|---|---|---|---|
| USC | Accelerometer, Gyroscope | 12 | People | Activity Detection | 38312 | 38.5 |
| AVE | Audio, Visual | 28 | Multiple Scenes | Vision | 8228 | 8601.6 |
| MHAD | Accelerometer, Skeleton | 11 | People | Activity Detection | 3956 | 187 |
| CGM | Colored image, Gray image | 10 | Digital | Vision | 160000 | 1293.8 |
| FLASH | GPS, LiDar, Camera | 64 | Traffic Scenes | Autopilot | 32923 | 5232.64 |
| ADM | Audio, Radar, Depth image | 11 | People | Medical | 22452 | 30208 |

where $\oplus$ represents the concatenation operation, and $\psi_{New}^{m_i}(\omega^{m_i}, x^{m_i})$ is the enhanced feature obtained by concatenating the real part feature and the imaginary part feature. It should be noted that the dimensions of $\psi_{New}^{m_i}(\omega^{m_i}, x^{m_i})$ remain the same as the original features $\psi^{m_i}(\omega^{m_i}, x^{m_i})$ to maintain the model structure and reduce computational costs.

## 4.2 Modal Rebalance via Multi-modal Constrained Loss

By incorporating complex space feature mapping and enhancement, it refined feature representation facilitating collaborative training for each modality and establishing a mutually beneficial relationship. This cooperative approach mitigates the impact of modal imbalance to some extent. However, we continue to face the challenge of optimization direction drift, posing an obstacle to further enhancing the model's performance.

To tackle the aforementioned challenge, we propose the **Multimodal Constrained Loss (MMCLoss)** to aid in adjusting the training direction of each modality. To implement this loss, we initially integrate the concept of phase, leveraging the distinctive attributes of the complex space. Based on Eq. (3) and Eq. (8), the phase $\varphi^{m_i}$ of modality $m_i$ is redefined as:

$$\varphi^{m_i} = \text{atan2}(\hat{\psi}_p^{m_i}(\omega^{m_i}, x^{m_i}), \hat{\psi}_r^{m_i}(\omega^{m_i}, x^{m_i})). \tag{11}$$

**Insight 2.** Utilizing polar coordinate space allows for a deeper understanding of complex feature information and a clearer perception of feature vector directions compared to rectangular space. Here, we elaborate on how phase information can guide model optimization directions. Initially, we calculate the phase $\varphi^{m_i}$ of each modality $m_i$ using Eq. (11), which encapsulates the feature phase information of modality $m_i$. Moreover, as illustrated in Eq. (3), the eigenphase represents the direction indicated by the eigenvector in polar space. In the multi-modal training, we observe that the phase and optimization direction of each modality undergoes constant changes with increasing training epochs. Based on this observation, we contend that modal phase information can be leveraged to dynamically adjust the direction of different modality optimizations.

Building on the prior discussion, in multi-modal learning, the dynamic phases act as indicators of the model's adaptive adjustments and optimization trajectory. To mitigate the performance degradation caused by drift in modal optimization directions, our objective is to align the optimization directions of each modality as closely as possible during training. To realize this objective, we introduce cosine similarity as a mechanism to constrain the feature phases:

$$\mathcal{L}_{\text{Cosine-similarity}}^{m_i} = 1 - \frac{\varphi^{m_i} \cdot \varphi^{m_i+1}}{\|\varphi^{m_i}\|\|\varphi^{m_i+1}\|}, \tag{12}$$

where we use $\varphi^{m_i}$ and $\varphi^{m_i+1}$ to represent the phase of the two modalities for calculating $\mathcal{L}_{\text{Cosine-similarity}}^{m_i}$. Through Eq. (12), we

keep the optimization directions of modality $m_i$ and mode $m_{i+1}$ gradually closer during the training process of multi-modal learning. However, how to ensure the correctness of the constrained optimization direction and maintain the performance of each modality is another serious challenge.

To this end, we propose a simple and efficient method in MM-CLosss. Specifically, we cleverly use cross-entropy loss to calculate the loss of logits of each modality:

$$\mathcal{L}_{CE}^{m_i} = -\frac{1}{N} \sum_{i=1}^{N} \log \frac{e^{f(x_i^{m_i})_{y_i}}}{\sum_{k=1}^{C} e^{f(x_i^{m_i})_k}}, \tag{13}$$

where $\mathcal{L}_{CE}^{m_i}$ represents the cross-entropy loss of modality $m_i$. Through Eq. (12) (13), we form the MMCLoss as follows:

$$\begin{aligned}
\mathcal{L}_{\text{MMC}} &= \sum_{m_i=0}^{M-1} \mathcal{L}_{\text{Cosine}-similarity}^{m_i} + \sum_{m_i=0}^{M} \mathcal{L}_{CE}^{m_i} \\
&= \sum_{m_i=0}^{M-1} (1 - \frac{\varphi^{m_i} \cdot \varphi^{m_i+1}}{\|\varphi^{m_i}\|\|\varphi^{m_i+1}\|}) \\
&\quad - \frac{1}{N} \sum_{m_i=0}^{M} \sum_{i=1}^{N} \log \frac{e^{f(x_i^{m_i})_{y_i}}}{\sum_{k=1}^{C} e^{f(x_i^{m_i})_k}}.
\end{aligned} \tag{14}$$

**Discussion.** The MMCLoss includes two components: cosine similarity and cross-entropy. It is crucial to note that the cosine similarity part of MMCLoss differs from previous approaches [9, 68] that concentrated on constraining the angle of the gradient. In our design, we leverage the complex phase to constrain the optimization direction between different modalities. Additionally, MMCLoss incorporates the cross-entropy loss of each modality, ensuring that the training process of different modalities consistently follows the correct path, thereby fostering cooperation between modalities while avoiding biased optimization directions. In this manner, MMCLoss facilitates the fusion model in achieving higher performance levels and maintaining modality balance from two perspectives.

Furthermore, we offer additional elucidation from the standpoint of visualization and complex triangle inequality theory in the full version to support the conclusion that the ERL-MR strategy enhances correlation and complementarity among diverse modality data, effectively governing the optimization speed and direction of the model.

## 4.3 Complexity Analysis

We present a time complexity analysis of the proposed ERL-MR strategy. Furthermore, the complete algorithm for the ERL-MR strategy is shown in Algorithm 1 in the full version. For simplicity, we assume that a convolutional neural network is applied in the feature extraction process. In addition to the computation complexity of both convolution operation per sample and the gradient update, the complexity for Euler feature transformation and feature compression can be calculated as $O(1) + O(2\zeta^2)$ where $\zeta$ is the dimensionality of $\psi_r^{m_i}(\omega^{m_i}, x^{m_{m_i}})$. For MMCLoss, the complexity of this operation is $O(M) + O(M \cdot N \cdot C)$ due to the $\mathcal{L}_{\text{Cosine-similarity}}$ and $\mathcal{L}_{CE}$ involved. Therefore, the introduced additional complexity for the proposed ERL-MR strategy is $O(2\zeta^2 + M + M \cdot N \cdot C)$.

## 5 Evaluation

### 5.1 Experimental Settings

To evaluate the performance of our ERL-MR strategy, we conduct extensive experiments on six benchmarking datasets. All experiments are developed using Python 3.9 and PyTorch 1.12 and evaluated on a server with an NVIDIA A100 GPU.

**Datasets.** We validated the ERL-MR strategy from **multiple task perspectives** using six multi-modal multimedia datasets in the experiment, including *USC Dataset* [73], *AVE Dataset* [51], *MHAD Dataset, Colored-gray MNIST Dataset (CGM)* [21], *FLASH Dataset* [45] and *Alzheimer's Disease Monitoring (ADM) Dataset* [37]. Table 1 summarizes the attribute information of the various multi-modal benchmark datasets in the experiment, and the complete information is provided in the full version.

**Models.** For the ADM dataset, we utilize the TDNN architecture [55] to extract audio features, while CNN layers are employed for extracting radar and depth image features. For USC, MHAD, and Flash datasets, we utilize a 2D-CNN model for the GPS data, whereas a 3D-CNN architecture is used to handle the skeleton, Lidar, and image data. For the AVE dataset, we adopt ResNet18 [12] as the encoder backbone and map the input data into a 512-dimensional vector. In terms of the audio modality, the data is converted into a spectrogram of size 257×1004. For the visual modality, we randomly select 4 frames from the video clips to construct the training dataset. For the CGM dataset, we design a model with 4 convolutional layers and 1 average pooling layer as the encoder.

**Hyperparameters.** For the ADM dataset, the learning rate is 1e-3, and the batch size is 64. Regarding the USC dataset, the learning rate is 1e-6, and the batch size is 16. For the USC, MHAD, and Flash datasets, the learning rate is 1e-3, and the batch sizes are 16, 16, and 96, respectively. For the AVE dataset and the CGM dataset, we set the learning rates to 1e-4 and 1e-3 respectively and the batch size is set to 64. In this experiment, we employ the SGD optimizer [44] with a momentum of 0.9 and a weight decay of 1e-4.

**Baselines.** To verify the proposed ERL-MR strategy, We adopt the following state-of-the-art "modality imbalance" solutions as the baselines: *PMR* [9], *MMCosine* [68], *OGM-GE* [40] and *Gradient-Blending* [60]. The complete baselines information is provided in the full version. To ensure fairness, we make efforts to reproduce all baseline methods and utilize the same network structure, number of training epochs, and SGD optimizer in all tasks.

### 5.2 Numerical Results

In this section, We have provided relevant experimental results on multiple research questions (**RQ1-RQ6**) related to the ERL-MR strategy on six multimodal datasets.

**Overall Performance of ERL-MR with Different Conventional Fusion Methods (RQ1).** We first apply the ERL-MR to several conventional fusion methods, including Concatenation, Summation, FiLM [41], and Gated [20], and subsequently evaluate their performance on multiple datasets, as presented in Table 2. The table also includes the performance results of each uni-modality. The results reveal significant variability in the performance of each uni-modal model across different datasets, indicating a common phenomenon of performance imbalance in multi-modal learning.

**Table 2: Performance Accuracy results on AVE, USC , MHAD and CGM datasets with various fusion methods. $^\dagger$ indicates the ERL-MR strategy is applied. The best results are underlined (also applicable throughout the text).**

| Dataset | AVE | USC | MHAD | CGM |
|---|---|---|---|---|
| **Fusion** | Acc (%) | | | |
| **Uni-Modal 1** | 38.1 | 56.0 | 84.9 | 99.3 |
| **Uni-Modal 2** | 11.9 | 46.0 | 58.8 | 60.4 |
| **Concatenation** | 42.0 | 61.0 | 94.6 | 58.4 |
| **Summation** | 40.0 | 62.6 | 94.9 | 59.1 |
| **Film** | 45.5 | 63.6 | 95.2 | 60.0 |
| **Gated** | 39.3 | 62.3 | 94.1 | 59.8 |
| **Concatenation**$^\dagger$ | 51.0 | 67.0 | 95.4 | 97.1 |
| **Summation**$^\dagger$ | 50.0 | 65.4 | 95.1 | 97.4 |
| **Film**$^\dagger$ | 49.3 | 67.1 | 95.8 | 92.6 |
| **Gated**$^\dagger$ | 52.7 | 66.1 | 96.0 | 76.4 |

It is noteworthy that in certain cases, the performance of the uni-modal models outperformed that of the multi-modal models employing the standard fusion methods. This finding, exemplified by Sec. 3.1, clearly illustrates the inhibitory relationship among modalities. Furthermore, based on experimental results, we find that the performance impact of dataset size on the ERL-MR strategy is obvious. The smaller datasets inherently possess limited feature information, resulting in relatively minor small performance enhancements. Conversely, the larger datasets exhibit a more substantial improvement effect. For example, ERL-MR strategy has yielded remarkable results in the CGM dataset. The multi-modal model leveraging this strategy surpasses the performance of the top-performing uni-modal model and overcomes the suppressive effect between modalities. In general, the application of our ERL-MR strategy leads to significant improvements on four datasets in comparison to each fusion method.

**Performance Comparison with State-of-the-art Baselines (RQ2).** We conduct a comparative analysis of our ERL-MR strategy against four state-of-the-art imbalanced modality modulation schemes: Gradient-Blending, OGM-GE, MMCosine, and PMR. For a comprehensive and fair comparison, in this experiment, we also test the performance of baselines using different fusion methods. As shown in Table 3, the experimental results show that the baseline methods outperform conventional fusion methods, yet they are still significantly surpassed by our proposed ERL-MR strategy. The main factor contributing to this improvement is the design of our *Euler Transformation*, which enhances the representation capabilities of various modalities, enabling them to reinforce mutually. When combined with the MMCloss, the ERL-MR strategy achieves even more substantial performance gains, benefiting from as the effective constraints ensure the accurate model optimization. Furthermore, it is worth mentioning that Gradient-Blending necessitates the training of an additional uni-modal classifier and thus involves additional computations to validate the results. OGM-GE can only be directly applied in the form of concatenation and is not applicable to not in FiLM, thereby necessitating periodic modulation during the training process. PMR requires additional calculations for the prototype,

**Table 3: Accuracy performance (%) comparison results between ERL-MR and four state-of-the-art baseline fusion methods on AVE, USC, MHAD, and CGM datasets.**

| Method | | G-Blend | | | | OGM-GE | | | | MMCosine | | | | PMR | | | | Ours | | | |
|---|---|---|---|---|---|---|---|---|---|---|---|---|---|---|---|---|---|---|---|---|---|
| Dataset | Modality | Concat | Sum | Film | Gated | Concat | Sum | Film | Gated | Concat | Sum | Film | Gated | Concat | Sum | Film | Gated | Concat | Sum | Film | Gated |
| USC | Accelerometer | 60.7 | 59.5 | 62.7 | 58.6 | 39.9 | 39.4 | 56.6 | 47.6 | 42.3 | 39.0 | 56.2 | 37.3 | 39.1 | 39.7 | 45.4 | 49.1 | 64.6 | 64.0 | 62.6 | 64.7 |
| | Gyroscope | 58.5 | 57.3 | 60.6 | 59.3 | 46.6 | 45.4 | 50.7 | 49.7 | 36.6 | 34.9 | 46.4 | 37.7 | 38.6 | 37.3 | 33.9 | 39.8 | 61.4 | 62.4 | 62.4 | 64.1 |
| | Fusion | 61.7 | 60.1 | 64.1 | 61.6 | 59.7 | 61.6 | 63.4 | 61.1 | 62.7 | 63.0 | 61.4 | 62.7 | 55.1 | 61.0 | 57.4 | 63.7 | 67.0 | 65.4 | 67.1 | 66.1 |
| AVE | Audio | 35.3 | 35.6 | 31.8 | 35.1 | 36.3 | 35.6 | 34.2 | 32.3 | 43.5 | 43.8 | 35.8 | 42.7 | 35.1 | 40.3 | 38.6 | 35.8 | 44.0 | 40.3 | 37.5 | 39.6 |
| | Visual | 24.1 | 27.6 | 24.4 | 25.1 | 12.2 | 13.2 | 12.1 | 11.8 | 16.2 | 20.9 | 10.2 | 10.4 | 16.7 | 17.4 | 16.9 | 16.4 | 35.1 | 30.6 | 25.1 | 32.1 |
| | Fusion | 45.0 | 49.8 | 38.3 | 44.5 | 41.3 | 44.5 | 42.0 | 35.8 | 42.3 | 48.8 | 39.1 | 46.3 | 42.0 | 43.3 | 40.5 | 40.5 | 51.0 | 50.0 | 49.3 | 52.7 |
| MHAD | Accelerometer | 81.3 | 82.7 | 91.8 | 82.2 | 45.1 | 54.3 | 85.5 | 81.7 | 65.7 | 67.6 | 90.7 | 61.7 | 54.3 | 39.4 | 46.5 | 24.1 | 82.6 | 81.5 | 93.2 | 79.4 |
| | Skeleton | 91.8 | 92.2 | 91.8 | 92.3 | 92.2 | 89.3 | 58.6 | 93.0 | 91.1 | 92.3 | 77.3 | 91.7 | 54.0 | 51.2 | 57.5 | 65.8 | 93.2 | 94.8 | 94.5 | 94.8 |
| | Fusion | 92.8 | 93.1 | 91.9 | 92.8 | 94.8 | 94.8 | 93.0 | 94.4 | 93.5 | 94.2 | 93.8 | 95.0 | 94.8 | 95.2 | 91.9 | 94.8 | 95.4 | 95.1 | 95.8 | 96.0 |
| CGM | Colored image | 99.1 | 99.2 | 99.1 | 99.3 | 21.3 | 19.6 | 22.0 | 11.3 | 98.7 | 98.8 | 99.2 | 99.1 | 91.8 | 92.1 | 91.7 | 92.2 | 99.4 | 99.5 | 99.5 | 99.5 |
| | Gray image | 63.0 | 63.6 | 63.2 | 63.6 | 57.1 | 59.2 | 10.4 | 66.8 | 23.0 | 20.6 | 14.0 | 14.0 | 65.8 | 62.8 | 61.2 | 63.1 | 62.8 | 65.0 | 69.1 | 68.7 |
| | Fusion | 92.8 | 94.1 | 74.3 | 76.1 | 69.1 | 64.1 | 66.7 | 66.7 | 68.5 | 65.9 | 81.2 | 81.2 | 77.2 | 78.5 | 76.3 | 68.7 | 97.1 | 87.4 | 92.6 | 76.4 |

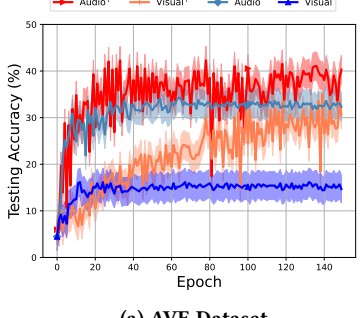

**(a) AVE Dataset**

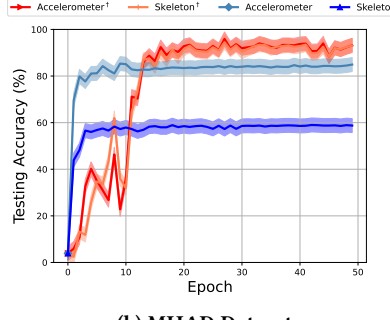

**(b) MHAD Dataset**

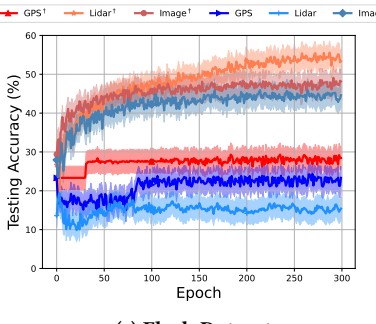

**(c) Flash Dataset**

**Figure 2: Performance curves of modality balance on AVE, MHAD, and Flash datasets. Note the summation fusion method used in the AVE and Flash datasets, while FiLM is used in MHAD. † indicates ERL-MR strategy is applied.**

resulting in significant computational overhead. In contrast, our method merely requires the implementation of a straightforward Euler component and is independent of both the fusion method and the network structure. *This enables our approach to be applicable to a wider range of scenarios.*

**Modality Imbalance Mitigation Performance Demonstration (RQ3).** In this experiment, we demonstrate the improvements in modality balance performance achieved by the ERL-MR strategy. The experimental results are depcited in Fig. 2. Specifically, Fig. 2a demonstrates the performance discrepancy between the visual and audio modalities in the AVE dataset. Without the applying ERL-MR strategy, the visual modality exhibited lower performance, while the audio modality demonstrated higher performance. With the applying ERL-MR strategy, however, we observed a continuous improvement in the performance of the visual modality during training, eventually reaching a performance level similar to that of the audio modality. In addition, we can find that the performance of the audio modality is further improved with the help of ERL-MR. Both modalities outperform the results obtained without ERL-MR, demonstrating a relatively balanced performance. This indicates that ERL-MR alleviates the modality imbalance and achieves the better uni-modal performance in this case. Likewise, the similar results are also observed on the MHAD dataset, providing additional

support for our assertion, as illustrated in Fig. 2b. To explore the performance of ERL-MR in more than two modalities application scenarios, we conducted experiments on the Flash dataset. The results are shown in Fig. 2c. Upon applying ERL-MR, the performance enhancements across all modalities, albeit to a lesser extent in the GPS modality. We attribute this observation to two factors. First, the tasks involving more modalities (more than 2) tend to be more complex, resulting in leading to the increased task complexity and inter-modality connections. Second, GPS data often exhibits high randomness, making the information obtained less effective during training. Nonetheless, the results in Fig. 2c demonstrate the effectiveness of our ERL-MR strategy in balancing uni-modal performance, even in complex tasks involving multiple modalities.

**Performance under Diverse Real-world Tasks (RQ4).**

*Autonomous Driving.* In this experiment, we conduct experiments on the Flash dataset and present the experimental results in Fig. 3a. The results reveal a significant modality imbalance issue among the three modalities of the Flash dataset when ERL-MR strategy is not applied. In this scenario, the GPS and lidar modalities are underutilized, and the image modality dominates. In this context, the implementation of ERL-MR strategy not only enhances the performance of the fusion network but also leads to better performance of the lidar modality. Additionally, due to the inherent

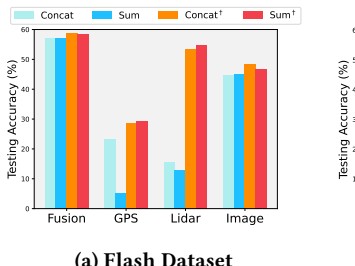

(a) Flash Dataset

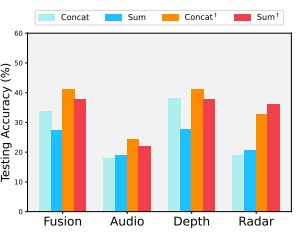

(b) ADM Dataset

**Figure 3: Experimental results are obtained by the concatenation and summation fusion methods on Flash and AD datasets.**

**Table 4: Performance on USC and AVE with the intermediate fusion methods (MMTM and CentralNet).**

| Method | MMTM | MMTM$^{\dagger}$ | CentralNet | CentralNet$^{\dagger}$ |
|---|---|---|---|---|
| **Dataset** | | | Acc (%) | |
| **USC** | 63.6 | **67.6** | 59.4 | **61.6** |
| **AVE** | 41.8 | **48.3** | 48.8 | **49.0** |

uncertainties in GPS modality data, traditional methods struggle to capture correlations between GPS modality and other modalities. In this work, ERL-MR strategy significantly enhances the performance of the GPS modality, aligning with its goal of promoting mutually beneficial cooperation rather than competition among different modalities. Highlighting the effectiveness of the ERL-MR strategy in the autonomous driving domain.

*Alzheimer's Disease Monitoring.* In this experiment, we conduct the ERL-MR strategy on ADM dataset. The experimental results are shown in Fig. 3b. We can find that the application of ERL-MR strategy shows the improvement in the accuracy of monitoring Alzheimer's disease. The training process reveals the dominance of the depth image modality, primarily owing to its direct responsiveness to patient behavior. In this context, the ERL-MR strategy significantly improves the performance of audio and radar modalities. Through the collaborative optimization of all three modalities, we achieve more accurate monitoring of Alzheimer's disease. This observation highlights the effectiveness of the ERL-MR strategy and its great potential in the healthcare domain.

**Impact of Different Fusion Stages on Performance (RQ5).** In the aforementioned previous experiments, we explore four fusion methods, where the fusion stage is positioned after either the encoder or the classifier. To further investigate the versatility of our ERL-MR strategy, we introduce incorporate two intermediate fusion methods, *i.e.*, MMTM [19] and CentralNet [54], testing on AVE and USC datasets. We analyze the performance of scenarios with ERL-MR and that of scenarios without ERL-MR. In the AVE dataset, we employ ResNet18 as the backbone and apply MMTM in the final three residual blocks. To simplify the experiment, we only use one frame per video. Due to the lightweight nature of the USC dataset, we only employ MMTM in the last layer of encoders. For CentralNet, we use it in each layer of encoders. The results in Table 4 demonstrate that our ERL-MR strategy achieves significant performance improvements when it is combined with intermediate fusion methods, that is, the fusion occurs at the encoder processing stage. The finding suggests that our proposed ERL-MR strategy is

**Table 5: Ablation experiments with multiple fusion methods conducted on the USC dataset. ✓ denotes the application of the method or constraint, while ✗ indicates its absence.**

| Dataset | USC | | | | | |
|---|---|---|---|---|---|---|
| **Fusion** | E C1 C2
✗ ✗ ✗ | E C1 C2
✓ ✗ ✗ | E C1 C2
✓ ✓ ✗ | E C1 C2
✓ ✗ ✓ | E C1 C2
✗ ✗ ✓ | E C1 C2
✓ ✓ ✓ |
| **Concat** | 61.0 | 63.4 | 63.7 | 64.8 | 61.4 | **67.0** |
| **Sum** | 62.6 | 65.3 | 64.0 | 64.3 | 61.9 | **65.4** |
| **FiLM** | 63.6 | 65.9 | 65.3 | 64.7 | 63.6 | **66.1** |
| **Gated** | 62.3 | 63.9 | 64.6 | 63.4 | 62.4 | **67.1** |

adaptable to various fusion stages, demonstrating good scalability and stability in diverse scenarios.

**Impact of Euler Transformation and MMCLoss on Performance (RQ6).** In the proposed ERL-MR strategy, the fundamental components are Euler transformation and MMCLoss. Therefore, we further analyze the impact of the two components on the overall performance of the fusion network. The accuracy results (%) are shown in Table 5, where **E** represents the Euler feature transformation method, **C1** and **C2** denote the cosine similarity constraint and cross-entropy constraint in MMCLoss, respectively. It is important to note that **C1** and **E** *are tightly coupled in our strategy*, thus the separate performance experiments using only **C1** were not conducted. The experimental results demonstrate that, except when only **C2** is applied, using each component individually or in combination results in performance improvements compared to the baseline method. This indicates an essential relationship between improving model performance and the inclusion of component **E**. Additionally, combining component **E** with either **C1** or **C2** results in limited improvement compared to applying component **E** alone. However, when all three components are simultaneously applied, the performance is further improved, providing additional evidence of the effectiveness of each component in the ERL-MR strategy. Specifically, component **E** provides fine-grained enhancement features for each modality, while **C1** and **C2** guide the overall optimization direction by constraining these enhancement features. This strategy addresses the issue of modality imbalance and bolsters the performance of multi-modal models.

## 6 Conclusion

In this work, we introduced the ERL-MR strategy as a novel approach to mitigate modality imbalance in multi-modal learning. This strategy effectively transforms the competitive dynamics between modalities, ensuring a stable feature optimization direction and promoting a mutually reinforcing collaboration among modalities. Importantly, this strategy is versatile, accommodating various multi-modal learning tasks with diverse data modalities and network architectures. Through comprehensive analysis and experimental validation across different datasets, encompassing various modalities, network structures, and application scenarios, the proposed strategy has consistently demonstrated its efficacy. ERL-MR emerges as a robust solution to address the challenges posed by modality imbalance in multi-modal learning, leading to significant improvements in model performance. These findings highlight the promising potential of the ERL-MR strategy for advancing robust multi-modal learning in real-world application scenarios.

## Acknowledgments

This work is supported by the Fundamental Research Funds for Heilongjiang Universities, China (Grant No. 2023-KYYWF-1449), in part by National Nature Science Foundation of China under Grant 62202398, by Guangdong Basic and Applied Basic Research Foundation under Grant 2023A151514 0137, and by Guangdong-Hong Kong Joint Laboratory for Data Security and Privacy Preserving (Grant No. 2023B1212120007)

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
