# OpenReview forum: "ERL-MR: Harnessing the Power of Euler Feature Representations for Balanced Multi-modal Learning"
_acmmm.org/ACMMM/2024/Conference — MM2024 Poster_

### Official Review · Reviewer_PX19 · 2024-05-12

**Rating:** 4
**Confidence:** 3

**Summary:**

This work addresses the issue of modal imbalance in multi-modal learning and proposes a mutually reinforcing learning framework using Euler representation learning. Extensive experiments have demonstrated the effectiveness of this method.

**Strengths:**

##  **This is an easy to understand work.**
1. The motivation is concise and reasonable, and the relevant work summary is substantial and rich.
2. The method statement is specific and appropriate, and theoretical analysis provides theoretical support for this work.
3. Abundant experiments have demonstrated the effectiveness of the proposed method.

**Limitations:**

## **Some minor concerns.**
**This is a good job, and the method of presenting the entire text is natural and appropriate. I understand most of the content, and I think this is a good job, and I only have a few small confusions about the experimental part. If the author's clarification can help me understand these confusions, I am willing to improve my rating.**

**Some missing implementation details:** For fusion operations Concatenation, Summation, FiLM, and Gated in Table 2 and Table 3, the authors did not explain how they were integrated with the proposed framework, or in which part of Figure 1 these operations were inserted. For example, adding fusion operations after FC, according to my understanding.

**Clarify some abnormal phenomena:** I understand that this work is similar to balancing the quality between different modalities. However, some counterintuitive results still require further clarification from the authors. For example, in terms of the reader's intuitive perception, after balancing like this work, the fusion effect should be even more outstanding due to the mutual promotion of learning effectiveness. However, on the CGM dataset, the fused effect does not seem to be better than the single-modal effect, at least as shown in Table 3. The effect of Colored image is much better than Fusion in most cases, and the authors need to explain this phenomenon.

**Further experimental supplement on Flash and ADM datasets:** The authors should present a figure for ADM dataset, which is same as Flash dataset Figure 2(c). Moreover, The authors should also add a performance comparison among G-Blend, OGM-GE, MMCosine, and PMR methods on Flash and ADM datasets. Otherwise, this part of the content is to some extent missing. If this part of the experiment cannot be added, please also explain the reason.

**Prospects for future work:** It is appropriate to add some prospects for future work in the conclusion section to help readers grasp the improvement direction of this work in the future.

**Suitability:**

3

---

### Official Review · Reviewer_DLXj · 2024-05-23

**Rating:** 3
**Confidence:** 4

**Summary:**

This paper proposes an Euler Representation Learning-based Modality Rebalance (ERL-MR) strategy, which reshapes the underlying com- petitive relationships between modalities into mutually reinforcing win-win situations while maintaining stable feature optimization directions. Specifically,

**Strengths:**

The motivation is well-supported and reasonable.
The experiments is sufficient, including various datasets, tasks, and modalities
The idea to transform the vector feature representations of diverse modalities into a complex vector space is intritesting

**Limitations:**

It would be better to add more detail in Figure 1, such as the notation of encoders, and modality, which would help readers to understand the paper better.

The derivation of formula 5 is hard to understand.  Why are the real and imaginary parts expressed like this? what's the parameters u_j?

Why the sine and cosine operations inherent in Euler’s formula can alleviate feature differences between different modalities, transforming the competitive relationship between modalities into a cooperative one. It seems not intuitive

The reviewer concerns about the reliability of the experiments. For the AVE dataset, the performance is far lower than that in existing paper, including OGM, and PMR, with the same training parameters. Besides, other widely used datasets, like Kinects-sound, CREMA-D are not considered

In general, the reviewer appreciates the motivation of this paper,  the logic of the method writing and the reliability of the experiment do not convince me.

**Suitability:**

3

---

### Official Review · Reviewer_excv · 2024-05-25

**Rating:** 4
**Confidence:** 4

**Summary:**

This paper proposes a novel strategy to address the issue of modality imbalance. The authors introduce the Euler Representation Learning-based Modality Rebalance (ERL-MR) strategy, which uses Euler’s formula to map features into a complex space, enhancing each modality and transforming competitive relationships into mutually reinforcing ones. Additionally, they propose a Multi-Modal Constrained (MMC) loss function that combines cosine similarity constraints of complex feature phases with cross-entropy loss to guide the optimization direction of the fusion network. Extensive experiments on six multi-modal datasets show that ERL-MR significantly outperforms state-of-the-art methods, achieving better modality balance and improved performance.

**Strengths:**

- This paper introduces a new approach by leveraging Euler’s formula to map features into a complex space, transforming competitive modality relationships into mutually reinforcing ones.
- Conducts extensive experiments on six real-world multi-modal datasets, demonstrating the practical applicability and effectiveness of the proposed method.

**Limitations:**

- The writing is quite poor, it is not clear enough to introduce EFT in detail, and how this relates to MMCLoss. E.g., the in eq. 4 what these two equations mean? How to learn the mapping? Besides, figure 1 is not clear, how the new features transform, and what is the meaning of the symbol after it.
- The coverage of related work is insufficient. For instance, the use of Euler's formula to transform features is not a novel concept, and it is crucial to cite the original references and the starting points in ML and multi-modal.
- Given the description it is hard to reproduce the results, it will be better to have related codes.
- It is not clear to me if the EFT is a technique especially suitable for multimodal applications, or it can also work in a single modality.
- The transformation of features into a complex space might reduce the interpretability of the model’s internal workings, making it harder to understand how individual modalities contribute to the final decision-making process.
- The Multi-Modal Constrained (MMC) loss function, while theoretically sound, could introduce additional overhead in the training process, potentially leading to longer training times compared to simpler baseline methods. It will be better to have experiments to verify this.

**Suitability:**

3

---

### Official Review · Reviewer_QKz1 · 2024-05-26

**Rating:** 5
**Confidence:** 3

**Summary:**

In this paper, the authors propose a Euler Representation Learning-based Modality Rebalance (ERL-MR) strategy, which reshapes the underlying competitive relationships between modalities into mutually reinforcing win-win situations. It is achieved by employing Euler’s formula to map features to complex space. The authors also introduce the multi-modal constrained loss based on cosine similarity to guide the optimization direction of the fusion network. By leveraging these approaches, the authors are able to solve the problem if modality imbalance, which impedes multi-modal learning from fully taking advantage of different modalities. Extensive experiments are conducted on several real-world datasets. And the paper is able to outperform the previous baselines.

**Strengths:**

+ Well-written article with clear problem formation
+ Detailed analysis and proofs are included.
+ Well-designed experiments and high performance over previous works. Ablation studies are also applied.

**Limitations:**

Weakness:
-Introducing existing solutions to the multi-modal area
-Less convincing baseline choices

Questions
I enjoyed reading this paper because it is easy for the reader to understand. The authors aim to solve the practical problem of modality imbalance in multi-modal computing. I also agree that the Euler feature representation method would be more effective. The experiments are well designed and convincing. However, I still have some questions about this paper.

The first question is about the Euler Feature Transformation. As introduced in Section 2.3, researches have been using Euler feature representations to capture the inherent geometric characteristics of the data. What is the major difference between this work and the previous works? The author claims that they are inspired by the Euler’s formulation. Do they simply migrate existing solutions to this new problem? Or the major contribution of this section is to build the complex mapping space?

The second question is about the experimental baselines. In Table 2, it seems that the authors directly use some simple models to serve as the baselines, i.e. Resnet 18 and simple 4-convolution, 1 average polling encoders. The multi-modal implementations of CGM are even far less effective than the uni-modals, which probably do not come from any articles. If SOTA baselines are applied, can this work still achieve similar performance? Similarly in Table 3, it would be more convincing if the authors could also compare the performance of their work and the SOTA models instead of only comparing with the SOTA “modality imbalance” solutions.

Minor issues:
Some tables and plots are too small and hard to read, such as Table 1, Figure 2 and 3.

**Suitability:**

3

---

### Meta-Review · Area_Chair_3xxT · 2024-07-05

**Recommendation:** Accept (Poster)
**Confidence:** 4

**Metareview:**

The study introduces a new approach to addressing modality imbalance in multi-modal learning by employing Euler’s formula to map features into a complex space, thus transforming competitive relationships into mutually reinforcing ones. Reviewers praised the detailed analysis, and robust experimental validation, demonstrating performance improvements over existing methods. Concerns were raised about the originality and detailed explanation of the Euler Feature Transformation, the choice of experimental baselines, and some clarity issues in the presentation. The authors' rebuttal to some extent addressed these concerns, providing necessary clarifications and additional experiments, which strengthened the overall understanding and robustness of the work. Given the responses, the reviewers reached a consensus on recommending acceptance. The authors are encouraged to revise the paper according to the reviewers' feedback to further enhance its clarity and quality.